# Berberine-Encapsulated Poly(lactic-co-glycolic acid)–Hydroxyapatite (PLGA/HA) Microspheres Synergistically Promote Bone Regeneration with DOPA-IGF-1 via the IGF-1R/PI3K/AKT/mTOR Pathway

**DOI:** 10.3390/ijms242015403

**Published:** 2023-10-20

**Authors:** Li Chen, Meng Tian, Jing Yang, Zhenxu Wu

**Affiliations:** 1Key Laboratory of Molecular Epigenetics of Ministry of Education, Institute of Cytology and Genetics, Northeast Normal University, 5268 Renmin Street, Changchun 130024, China; chenl166@nenu.edu.cn (L.C.); tianm132@nenu.edu.cn (M.T.); yangj357@nenu.edu.cn (J.Y.); 2Key Laboratory of Polymer Ecomaterials, Changchun Institute of Applied Chemistry, Chinese Academy of Sciences, 5625 Renmin Street, Changchun 130022, China

**Keywords:** polymer microspheres, berberine (BBR), IGF-1, bone regeneration, osteogenic differentiation

## Abstract

Polymer microspheres have recently shown outstanding potential for bone tissue engineering due to their large specific surface area, good porosity, injectable property, good biocompatibility, and biodegradability. Their good load-release function and surface modifiability make them useful as a carrier of drugs or growth factors for the repair of bone defects in irregularly injured or complex microenvironments, such as skull defects. In this study, berberine (BBR)-encapsulated poly(lactic-co-glycolic acid) (PLGA)/hydroxyapatite (HA) microspheres were fabricated using electrified liquid jets and a phase-separation technique, followed by modification with the 3,4-hydroxyphenalyalanine-containing recombinant insulin-like growth–factor-1 (DOPA-IGF-1). Both the BBR and the IGF-1 exhibited sustained release from the IGF-1@PLGA/HA-BBR microspheres, and the composite microspheres exhibited good biocompatibility. The results of the alkaline phosphatase (ALP) activity assays showed that the BBR and IGF-1 in the composite microspheres synergistically promoted the osteogenic differentiation of MC3T3-E1 cells. Furthermore, it was confirmed that immobilized IGF-1 enhances the mRNA expression of an osteogenic-related extracellular matrix and that BBR accelerates the mRNA expression of IGF-1-mediated osteogenic differentiation and cell mineralization. Further cellular studies demonstrate that IGF-1 could further synergistically activate the IGF-1R/PI3K/AKT/mTOR pathway using BBR, thereby enhancing IGF-1-mediated osteogenesis. Rat calvarial defect repair experiments show that IGF-1@PLGA/HA-BBR microspheres can effectively promote the complete bony connection required to cover the defect site and enhance bone defect repair. These findings suggest that IGF-1@PLGA/HA-BBR composite microspheres show a great potential for bone regeneration.

## 1. Introduction

Bone tissue engineering has attracted great interest in the regeneration of bone defects caused by trauma, tumors, osteoporosis, etc. Recently, microspheres have displayed an outstanding potential in bone tissue engineering due to their advantages, such as a wide specific surface area, good porosity, biocompatibility, biodegradability, and load-sustained release function [1]. It is believed that treatment with microspheres can be recommended for irregular bone defects caused by congenital malformations, trauma injuries, infections, bone diseases (e.g., osteoporosis), and tumors. In the process of bone regeneration, microspheres maintain a certain three-dimensional configuration, providing a microenvironment for cell adhesion and guiding cell migration and growth [2,3]. Moreover, they can be used as carriers of growth factors or drugs that further regulate the formation of new bone [4].

Tissue engineering microspheres based on poly-L-lactic acid (PLA), poly(caprolactone-co-glycolide) (PCL), or poly(lactic-co-glycolic) acid (PLGA) have been widely employed in bone tissue regeneration with promising effects [5,6,7]. Among them, PLGA is an ideal matrix material for bone tissue engineering, approved by the U.S. Food and Drug Administration (FDA) [8,9]. However, the shortcomings of PLGA, such as a poor biological activity and acidic degradation by-products, cannot be ignored. It was found that nano-inorganic ceramic materials such as hydroxyapatite (HA) were blended with PLGA to improve the osteoconductive and osteoinductive abilities of the materials. Jiao et al. fabricated PLGA/HA microspheres via electrified liquid jets and a phase-separation technique and found that the composite microspheres were more osteogenic than the PLGA microspheres [10], indicating that PLGA/HA composite microspheres are a satisfactory basic biomaterial for bone tissue engineering, although the osteogenic activity of the composite microspheres can be further enhanced by the addition of active molecules.

Berberine (BBR), a kind of isoquinoline alkaloid extracted from coptis chinensis, has a wide range of pharmacological activity and clinical applicability, including antibacterial effects, in vivo glucose/cholesterol regulation, and immunomodulation through enhanced leukocyte phagocytosis [11]. It has been widely used in the treatment of gastroenteritis, bacillary dysentery, tuberculosis, scarlet fever, acute tonsillitis, and respiratory tract infections [12]. In vitro and in vivo investigations have additionally demonstrated its anti-arrhythmic and anti-tumor effects [12]. The activity of BBR in bone homeostasis has been in the spotlight due to some reports pointing to its involvement in bone resorption. Numerous studies have confirmed the negative effect of BBR on bone resorption. Chen et al. investigated the protective effect of berberine on senile osteoporosis in mice and preliminarily assessed its potential mechanism [13]. The results suggest that berberine exerts a potent bone-protective effect by inhibiting the accumulation of marrow adipocytes and bone resorption. And it is possible that this effect is achieved through the cAMP/PKA/CREB signaling pathway. Another study from Hu’s group reported the inhibitory effect of berberine on osteoclast formation and survival, induced by the receptor activator of the nuclear factor kappa B (NF-κB) ligand (RANKL) [14]. This finding suggests that berberine inhibits osteoclast formation and survival through the suppression of NF-κB pathway activation and that this pathway in the osteoclast lineage cells is highly sensitive to berberine treatment. In addition, a few studies have focused on the osteogenesis effect of BBR. Zhou’s group evaluated the modulation of BBR on the lipopolysaccharide (LPS)-mediated osteogenesis and adipogenesis imbalance in rat-bone-marrow-derived mesenchymal stem cells [15]. The results showed that BBR was able to increase the mRNA expression levels of the osteoblastic genes. In osteoblastic differentiated cells, a decreased ALP production after LPS treatment was reversed through BBR co-incubation. The action of BBR was attenuated by compound C, suggesting that the role of BBR may be partly due to the activation of AMP-activated protein kinase. Lee’s study demonstrated that BBR could enhance the expression of osteogenic marker genes, including osteopontin (OPN) and osteocalcin (OCN), and that it promotes the transcriptional activity of runt-related transcription factor 2 (RUNX2), a key osteogenic transcription factor [16]. Further mechanism studies showed that BBR activated the expression of p38 mitogen-activated protein kinase (MAPK) and cyclooxygenase 2 (COX2), thereby promoting RUNX2 activity and osteoblast differentiation. These findings demonstrate the potential of BBR as an osteogenic signaling molecule for bone tissue engineering.

IGF-1 has been shown to be an efficient growth factor associated with bone regeneration [17]. IGF-1 derivatives have been used to stimulate the bone healing process [18]. Lee et al. reported that IGF-1 treatment increased the expression of osteopontin, alkaline phosphatase, and osteocalcin in osteoblast cells at the defect site and promoted new bone formation at the skull defect site [19]. Choi’s study showed that Ti substrates loaded with IGF-1 and BMP-2 nanoparticles significantly promote attachment, proliferation, proliferation, and alkaline phosphatase activity in human adipose-derived stem cells (hADSCs) [20]. The effects of IGF-1 have been reported to be primarily achieved through the PIK3/AKT/mTOR pathway [21]. But the potency of IGF-1 is limited when applied in vitro and in vivo due to its short half-life and rapid release from the defect site. To overcome these shortcomings, a growing interest has arisen in the development of biomaterials with an immobilization of growth factors that would regulate specific cellular functions. Zhang et al. developed a three-dimensional porous PLGA/HA scaffold based on the dual delivery of bone morphogenic protein 2 (BMP-2) and IGF-1 via polydopamine (PDA) coating [22]. This study demonstrated that PDA-layer surface modification is more efficient than physical adsorption to immobilize BMP-2 and IGF-1 on the scaffold surface and that the immobilized growth factors are released slowly and steadily from the scaffold in a sustained manner. And the dual release of BMP-2 and IGF-1 could promote cell proliferation and osteogenesis differentiation. In our previous work, we incorporated a small pentapeptide tag consisting of Tyr-Lys-Tyr-Lys-Tyr (YKYKY) and a residue at the C-terminus of IGF-1, to convert this tag to DOPA-Lys-DOPA-Lys-DOPA using tyrosinase (DOPA-IGF-1) [23]. Recently, electroactive microspheres have been prepared by immobilizing an aniline tetramer (AT) on PLGA/HA microspheres. Subsequently, DOPA-IGF-1, inspired by bioorthogonal chemistry, was modified on the surface of electroactive microspheres [7]. It was found that DOPA-IGF-1 synergistically induces mineralization, osteogenic differentiation, and defect regeneration in rat skulls.

In the present study, BBR-encapsulated PLGA/HA microspheres were fabricated via electrified liquid jets and a phase-separation technique. The microspheres were then modified with DOPA-IGF-1. The loading contents of BBR and DOPA-IGF-1 were optimized. Afterward, the osteogenic activity of composite microspheres was evaluated in vitro on MC3T3-E1 cells. The synergistic effect of BBR and IGF-1 in osteogenesis was studied at the molecular level. Finally, the osteogenic repair capability of the composite microsphere was evaluated through repair experiments on a rat skull defect model (Figure 1). BBR combined with IGF-1 was innovatively used in this study to improve the osteogenic activity of bone tissue engineering composite microspheres. The synergistic effect of BBR and IGF-1 in promoting bone repair has been identified for the first time. This study provides a new insight into the efficient selection of biomolecules for bone tissue engineering.

## 2. Results and Discussion

### 2.1. BBR Encapsulating and Release

The encapsulation efficiency (EE), load capacity (LC), and release profile of the BBR-encapsulated microspheres were detected. The mass ratio of BBR in the solute during microsphere fabrication was set to 0.01 wt% (0.01BBR), 0.05 wt% (0.05BBR), 0.1 wt% (0.1BBR), 0.5 wt% (0.5BBR), and 1 wt% (1BBR), respectively. The results of the EE and LC harvested for the microspheres are shown in Figure 1A and Table 1. Both the EE and LC increased with the initial BBR contents.

The release behavior of BBR in different groups over 168 h is shown in Figure 1B and Table 2. Drugs in microspheres showed a rapid release in the early stage. However, the timing of the release curve into the plateau was different in each group. The 1BBR group reached the PR plateau after 24 h. The rapid release of BBR lasted for 48 h in the 0.01BBR and 0.1BBR groups, and 72 h in the 0.5BBR and 0.05BBR groups. After that, BBR began to release smoothly and slowly in these groups. The PR gradually increased as the drug content decreased.

According to the results, the EE of BBR in the 0.1BBR, 0.5BBR, and 1BBR groups was above 50%. And the EE of BBR in 0.5BBR and 1BBR reached 73.07 ± 3.56% and 85.77 ± 4.75%, respectively. It is worth noting that the EE of these two groups was much higher than that of composite microspheres fabricated by other methods, such as the solvent evaporation method (62.42 ± 0.55%) [24], membrane emulsification method (50–67%) [25], and emulsion chemical cross-linking method (65.2 ± 1.6%) [26]. It is speculated that when a BBR-containing droplet falls into the receiving solution, the surface of the microsphere solidifies rapidly, resulting in the rapid encapsulation of BBR in the microsphere. In addition, BBR has a lower solubility in the receiving solution, resulting in less drug loss. However, the EE decreased as the initial BBR content decreased. This may be caused by the dissolution of the BBR in the receiving solution. Stable drug release improves drug efficacy, extends action time, and reduces adverse reactions [27]. The results of this study showed that BBR in composite microspheres was rapidly released in the early or intermediate stages (0~96 h). The proportion of bursts released in groups with different BBR content was also different. As the BBR content of microspheres increased, the release rate gradually decreased. This may benefit from the proportion of BBR adhered to the surface of the microspheres and the solubility of BBR in PBS. BBR is slightly soluble in water, so the percentage of BBR released into PBS was higher in the low-concentration group and lower in the high-concentration group. The 0.01BBR group was released more completely than the other groups. This result is consistent with previous reports that the group with the highest concentration of the drug had the smallest percentage of release [27]. After the initial burst release, the drug entered a slow-release phase due to diffusion.

### 2.2. Effect of Different BBR Contents on MC3T3-E1 Cells

To determine the applied dosage of BBR, the proliferation and ALP activity of MC3T3-E1 cells planted on composite microspheres with different BBR content were detected. According to the results in Figure 1C,D, cells in all groups proliferated continuously with culture. The 0.01BBR, 0.05BBR, and 0.1BBR groups showed no obvious toxicity to cells, while the 0.5BBR and 1BBR groups significantly inhibited cell proliferation (*p* < 0.05). The data obtained at different time points consistently demonstrated a significant enhancement in cell proliferation for both the 0.01BBR group and the 0.05BBR group during culture (*p* < 0.05).

ALP activity is the key to evaluating osteogenic differentiation. The results of relative ALP activity are shown in Figure 1E,F. The BBR-encapsulated microspheres in each group were found to promote ALP activity in MC3T3-E1 cells on Days 3 and 7 (*p* < 0.05). However, the ALP activity was highest in the 0.1BBR group and significantly higher than in the control group (*p* < 0.05). Based on the results of the proliferation and ALP activity assays, 0.1BBR was selected for further investigation in this study and named PLGA/HA-BBR.

Due to its activity in osteogenesis [15,16], BBR has the potential to be used as an osteogenic signaling molecule for bone tissue engineering. It has been demonstrated that cell viability can be significantly reduced by BBR in a dose-dependent state of the subject [28]. In the present study, the optimal content of BBR was determined based on the results of the proliferation and ALP activity assays. This is due to the fact that microspheres of 0.1BBR not only induce the osteogenic differentiation of MC3T3-E1 cells to the greatest extent, but are also not toxic to cells.

### 2.3. Adhesion and Release of DOPA-IGF-1

The adhesion and release of DOPA-IGF-1 were quantitatively detected via the ELISA method. According to the result in Appendix A, the adhesion rate of DOPA-IGF-1 on microspheres was 90.28 ± 0.18%. As shown in Appendix A, DOPA-IGF-1 on microspheres was released at an extremely slow rate over time. After 48 h, 93.19 ± 1.17% of DOPA-IGF-1 remained attached to the surface of microspheres.

Due to the presence of bionic mussel protein residues, DOPA-IGF-1 is considered to have a universal adhesion force on the solid surface. In previous work, DOPA-IGF-1 has been demonstrated to form stable adhesions with Ti plates [23] and PLGA/HA microspheres [7]. This is consistent with the results of the present study.

### 2.4. Characterization of Microspheres

The general morphology, ESEM photographs, and particle size distribution of microspheres are shown in Figure 2. On the whole, the surface of the PLGA/HA microspheres showed a white color, while the BBR-encapsulated microspheres appeared a pale-yellow color due to the loaded BBR (Figure 2A). This indicated that BBR was successfully incorporated into the composite microspheres. It also demonstrated the excellent sphericity and homogeneity of the microspheres. All groups of the microspheres exhibited a good sphericity with particle size in the range of 300–450 μm (Figure 2B). The average diameter of the PLGA/HA, PLGA/HA-BBR, IGF-1@PLGA/HA, and IGF-1@PLGA/HA-BBR microspheres was comparable, being 412.47 ± 52.82, 408.33 ± 47.35, 404.34 ± 42.14, and 415.26 ± 41.75 μm, respectively. The microspheres in each group showed a good sphericity, reaching more than 0.95 (Appendix A). The surfaces of the microspheres in all groups were more or less rough. Of these, IGF-1@PLGA/HA-BBR microspheres have the roughest surfaces.

The FT-IR spectra of the microspheres are shown in Figure 2C. The stretching vibration absorption peak of the phenyl group, which indicates the existence of BBR, can be seen near 3322 cm^−1^, 2946 cm^−1^, and 2845 cm^−1^ in the groups of PLGA/HA-BBR and IGF-1@PLGA/HA-BBR. The peaks near 1588 cm^−1^ and 1652 cm^−1^ are the peaks of acylamino in DOPA-IGF-1, which can be observed in IGF-1@PLGA/HA and IGF-1@PLGA/HA-BBR groups. The peaks near 560 cm^−1^ and 605 cm^−1^ are the bending vibrational peaks of P–O, which represent the inorganic phosphate component of HA. The characteristic peaks at 1597 cm^−1^, 1032 cm^−1^, 827 cm^−1^ represent C=C, –C–O–C–, and –C–C–, respectively.

The electrified liquid jet and phase-separation methods are capable of producing uniformly sized, controllable composite microcarriers [10]. This technique was demonstrated to have two main significant advantages: (1) easy control over the shape and size of the microspheres, and (2) production of microcarriers with narrow size distributions and highly uniform morphology. In the present study, the PLGA/HA composite microspheres containing BBR were fabricated for the first time using electrified liquid jets and a phase-separation technique. By adjusting the preparation parameters, the size of the microspheres was successfully controlled between 300 and 450 μm. Microspheres of this size are more suitable for cell growth and bone tissue engineering applications [29]. The surface of the composite microsphere was roughened due to the incorporation of HA. This result was consistent with Jiao’s report [10]. The IGF-1@PLGA/HA-BBR microspheres exhibit the most pronounced surface roughness, potentially attributed to the exposure of IGF-1 and BBR on the microsphere’s surface. And a rougher surface would be more conducive to cell growth.

### 2.5. In Vitro Biocompatibility of Composite Microspheres

To evaluate the biocompatibility of composite microspheres, the adhesion and viability of MC3T3-E1 cells were detected. A FITC-staining assay was carried out and representative images are shown in Figure 3A. The number of cells marked green by FITC adhered to IGF-1@PLGA/HA-BBR microspheres was slightly higher than in the other groups. However, the cells on the microspheres in each group grew well. This result fully indicated that the amount of BBR released during the sustained drug release period (3 days) did not cause a significant negative effect on cell adhesion and growth. The results of cell viability (CCK-8) are shown in Figure 3B,C. The cells in all groups proliferated continuously with culture. At Days 3 and 7 of cell culture, the cell proliferation of MC3T3-E1 cells was significantly higher in all experimental groups compared to PLGA/HA and control groups (*p* < 0.05). In addition, cell viability at 7 days was significantly higher in IGF-1@PLGA/HA-BBR microspheres than in IGF-1@PLGA/HA and PLGA/HA-BBR microspheres (*p* < 0.05).

Good biocompatibility is a basic requirement for tissue engineering materials. PLGA/HA composites have been reported as mature bone repair materials with excellent biocompatibility. It has also been reported that HA in PLGA/HA composites significantly enhances cell adhesion and proliferation [30]. IGF-1 has been widely recognized as effective for cell viability, and DOPA-IGF-1 is thought to be superior to natural IGF-1 in terms of long-term activity. According to Zhang’s research, DOPA-IGF-1 modified on a Ti plate can significantly promote cell proliferation [23]. This was validated in our study. According to the present study, BBR does not inhibit the growth of MC3T3-E1 cells in either the early burst release phase (3 days) or the intermediate sustained release phase (7 days). Nevertheless, the IGF-1@PLGA/HA-BBR group has certain advantages in terms of biocompatibility compared to other groups.

### 2.6. In Vitro Osteogenesis of Composite Microspheres

The ability to induce osteogenic differentiation is important for evaluating the biological activity of bone repair materials. Levels of alkaline phosphatase activity can be used to characterize the early stage of osteogenic differentiation of cells. The staining and analysis results of ALP activity are shown in Appendix A and Figure 3D,E. On Day 3, all composite groups containing BBR or IGF-1 promoted ALP activity in MC3T3-E1 cells compared to the control group (*p* < 0.05). However, the ALP activity of cells in the IGF-1@PLGA/HA-BBR group was the highest, which was higher than that of the PLGA/HA group (*p* < 0.05). After 7 days of culture, ALP activity was significantly enhanced in all experimental groups compared to the control group (*p* < 0.05). The activity level of ALP in IGF-1@PLGA/HA-BBR was still the highest and significantly higher than those in the PLGA/HA and IGF-1@PLGA/HA groups (*p* < 0.05). The results showed that BBR and IGF-1 in composite microspheres could effectively promote the osteogenic differentiation of cells. Previous studies examined the differentiation of BBR-treated cells after approximately 7 days of culture and found a significant increase in ALP activity, suggesting that BBR can induce osteogenic differentiation of mesenchymal stem cells or MC3T3-E1 cells [15,16]. In addition, it has been shown that ALP activity in MC3T3-E1 cells is significantly increased by the addition of HA or IGF-1, revealing the additive effect of HA or IGF-1 in the early stages of osteogenesis [7,22]. Our results are in general agreement with previous reports, namely, after 7 days of culture, the ALP levels in 3T3 cells in all BBR and IGF-1 groups were significantly increased than those in the control group. And the synergistic effect of BBR and IGF-1 makes their inclusion in ALP activity even more significant. In addition, ALP levels in each group of cells at Day 3 gave us insight into the osteogenic differentiation of MC3T3-E1 cells promoted by BBR and IGF-1 early in the culture.

Figure 4A–D show the expression levels of mRNA associated with osteogenesis. Collagen I (COL1) is a major component of the bone extracellular matrix [31]. In the early stages of bone repair, osteoblasts colonize the defect and begin to secrete extracellular matrix, including COL1, to build the skeleton for osteoblast growth and mineralization. The large amount of early COL1 deposition will provide sufficient cell colonization and mineralization sites for bone repair. According to the results (Figure 4A), on Day 3, expression levels of COL1 in IGF-1-containing groups were significantly higher than those in the control group (*p* < 0.05). COL1 expression levels in PLGA/HA and PLGA/HA-BBR groups were higher than those in the control group, but only the difference between PLGA/HA-BBR and the control group was statistically significant. On Day 7, the expression levels of COL1 were extremely stimulated in IGF-1-containing groups compared to the other groups (*p* < 0.05). Meanwhile, the COL1 level in the PLGA/HA and PLGA/HA-BBR groups also significantly enhanced (*p* < 0.05). The expression of COL1 can be promoted by HA, BBR, and IGF-1 in MC3T3-E1 cells, with IGF-1 being the most effective factor.

The expression of RUNX2 is an important marker of the differentiation of osteoprogenitor cells into osteoblasts [32]. In RUNX2-mediated osteogenic differentiation, cells first transmit signals to the nucleus to direct the expression of RUNX2. After that, the cells undergo osteogenic differentiation under the intracellular action of RUNX2 and express other osteogenesis-related proteins (OCN, OPN, etc.). At Day 3, RUNX2 expression levels in the IGF-1 groups were significantly higher than in the control and PLGA/HA groups (*p* < 0.05). The expression of RUNX2 was also stimulated in the PLGA/HA-BBR group compared with control and PLGA/HA (Figure 4B). But only the difference between PLGA/HA-BBR and the control was significant (*p* < 0.05). After 7 days, RUNX2 expression levels in IGF-1 groups were still higher than in the other three groups. The differences between IGF-1@PLGA/HA-BBR and the three groups were significant (*p* < 0.05). Moreover, the difference between IGF-1@PLGA/HA and PLGA/HA was also significant (*p* < 0.05). The BBR-only group could also slightly stimulate the expression of RUNX2 on PLGA/HA microspheres. The difference, however, was not obvious. This indicated that IGF-1 is conducive to RUNX2 expression, and BBR can accelerate IGF-1-mediated RUNX2 expression.

Osteocalcin (OCN) plays an important role in regulating bone calcium metabolism and is a biochemical marker for bone metabolism. As shown in Figure 4C, there was little difference between the groups on Day 3, as the expression of OCN often occurs in the middle and late stages of culture. After 7 days, the IGF-1@PLGA/HA-BBR group showed an outstanding acceleration in OCN expression compared to the other groups (*p* < 0.05). IGF-1@PLGA/HA also showed a non-significant promoting effect compared with PLGA/HA. BBR promoted cell mineralization induced by IGF-1. Osteopontin (OPN) is a glycosylated protein widely distributed in the extracellular matrix. It is an important bone matrix protein that is closely related to bone formation and development [33]. Based on the present study, on Day 3, the expression level of OPN was much lower in the control group than in the other four groups (Figure 4D) (*p* < 0.05). OPN expression levels in the IGF-1@PLGA/HA-BBR and IGF-1@PLGA/HA groups were significantly higher than those in the PLGA/HA and control groups (*p* < 0.05). The OPN expression level in the IGF-1@PLGA/HA-BBR group was the highest. At Day 3, the OPN expression was significantly enhanced by BBR in the PLGA/HA microspheres (*p* < 0.05). But on Day 7, there was no significant difference between the PLGA/HA-BBR and PLGA/HA groups. OPN expression was mainly increased by the immobilized IGF-1, and the activity of BBR on the improvement of OPN expression should also be flagged. In total, the RT-PCR results indicated that immobilized IGF-1 can enhance mRNA expression in the extracellular matrix associated with osteogenesis. Gene expression involved in osteogenic differentiation and cell mineralization is also upregulated by IGF-1. And BBR can accelerate IGF-1-mediated osteogenic differentiation and cell mineralization.

### 2.7. Protein Expression by Western Blotting

To understand the mechanism by which BBR cooperates with IGF-1 to promote bone regeneration, MC3T3-E1 cells cultured on composite microspheres for 7 days were collected and IGF-1-mediated osteogenic differentiation pathway proteins were detected (Figure 4E). Levels of proteins in the IGF-1/IGF1R signaling pathways, such as IGF1R, p-AKT, and p-mTOR, were detected in different groups of cells. Remarkably, Western blot analysis showed increases in IGF1R, p-AKT, and p-mTOR levels in the BBR or IGF-modified microspheres, compared to the control or PLGA/HA groups. And levels of these proteins were most elevated in the IGF-1@PLGA/HA-BBR group (Figure 4F–H). These results suggested that both BBR and IGF-1 can activate the IGF1R/PI3K/AKT/mTOR pathway in MC3T3-E1 cells. The synergistic effect of BBR and IGF-1 can further increase the activation of this pathway, thereby accelerating the progression of downstream signaling pathways and leading to the expression of osteogenic genes.

The results confirmed the role of BBR-contained composites in promoting osteogenesis, which was further enhanced by IGF-1. However, the mechanism still needs to be studied. Hui Liu et al. elucidated the mechanism of action of Morroniside in preventing bone loss and found that Morroniside could promote osteoblast differentiation by activating the PI3K/AKT/mTOR pathway, which can be used to treat osteoporosis [34]. This study demonstrated that the PI3K/AKT/mTOR is an effective axis in osteogenic differentiation. In addition, mTORC1, a typical mTOR, has emerged as a common effector mediating the bone anabolic effects of IGF-1. It has been reported that IGF-1 stimulates the osteoblastic differentiation of mesenchymal stem cells (MSCs) by activating mTOR [21]. The role of BBR in the PI3K/AKT/mTOR pathway is controversial. Shi et al. investigated the anticancer effects of BBR on anaplastic thyroid carcinoma (ATC) cancer cell lines in vitro and found that the ratios of p-PI3K/PI3K and p-mTOR/mTOR in BBR-treated ATC cancer cells were lower, while the ratios of p-AKT/AKT were significantly increased [35]. In Sheng’s report, BBR administration significantly elevated the expression of p-AKT but reduced the phosphorylation of mTOR in attenuated hepatic injury after ischemia/reperfusion [36]. Song et al. explored the effect of different doses of BBR on the PI3K/AKT/mTOR pathway in carotid atherosclerosis. The findings revealed that BBR up-regulated the expression of p-PI3K and p-mTOR, but down-regulated the expression of p-AKT in a dose-dependent manner [37]. Obviously, the effect of BBR on the PI3K/AKT/mTOR pathway is variable in different tissues and cells at different doses. The present study indicated that the BBR could significantly up-regulate the expression of IGF-1R and significantly enhance the phosphorylation of PI3K/AKT/mTOR. Increased activation of this pathway is closely related to IGF-1-mediated osteogenic differentiation. BBR synergistically accelerates IGF-1-mediated osteogenic differentiation and cell mineralization via the IGF-1R/PI3K/AKT/mTOR pathway.

### 2.8. Repair Effect of Skull Injury

The changes in skull defects observed by micro-CT over time are shown in Figure 5, Table 3 and Table 4. The defects in each group were gradually covered by new bone tissue. The regeneration coverage rate of IGF-1@PLGA/HA-BBR was significantly higher than that of the control group and PLGA/HA group (*p* < 0.05) at all time points. The defect coverage rate also increased significantly in the PLGA/HA-BBR group compared to the control and PLGA/HA groups (*p* < 0.05) (Figure 5B). The regeneration coverage rate of the IGF-1@PLGA/HA group was lower than that of the PLGA/HA-BBR group at Day 40 but exceeded that of PLGA/HA-BBR at Day 60.

The IGF-1@PLGA/HA-BBR group showed a significantly higher bone volume over total volume (BV/TV) at the defect sites at each time point compared to the other groups (*p* < 0.05) (Figure 5C). The BV/TV in the PLGA/HA-BBR group and IGF-1@PLGA/HA group were consistently significantly higher than those in the control and PLGA/HA groups (*p* < 0.05). Similar to the regeneration coverage rate results, BV/TV in the PLGA/HA-BBR group was higher than that in the IGF-1@PLGA/HA group on Day 40, but lower than that in the IGF-1@PLGA/HA group on Day 60. These results indicated that the IGF-1@PLGA/HA-BBR could effectively promote the complete bony connection to cover the defect site.

H&E and Masson staining were used to highlight typical features in tissue sections. Figure 6A shows the results of H&E staining. Connective tissue was formed along or around the occupation of material. The occupied part of the newborn tissue grew and gradually covered the defect, while the complete osseous connection that acted as a “bone bridge” only formed over the defect in the IGF-1@PLGA/HA-BBR group. The BBR-containing microspheres were surrounded by denser connective tissue, which would provide sites for the mineralization of new bone. In particular, in the IGF-1@PLGA/HA-BBR group, the new bone formed thicker callus, and the repair effect was significantly better than that in the other groups. In all groups, the connective tissue around IGF-1@PLGA/HA-BBR microspheres was the densest, with a large amount of tissue growing into the interior of the microsphere. Figure 6B shows the results of Masson staining. The results showed that most of the defect sites in each group had been covered with newborn tissue. Scattered mature bone tissue stained red can also be observed inside the blue new bone. The thickness of the new bone in the BBR-containing groups was thicker than that in the other groups. And there are more red-stained mature bone tissues in the IGF-1-containing groups. In addition to the new callus observed at the base of the defect, blue-stained new bone tissue was also seen around the microspheres. Moreover, the new bone tissue surrounding the IGF-1@PLGA/HA-BBR microspheres was more numerous and denser than in the other groups. The results confirmed that IGF-1@PLGA/HA-BBR microspheres could significantly improve the effect of bone defect repair.

It has been shown that BBR negatively regulates bone absorption, promotes the expression of osteogenic genes, and has protective effects against senile osteoporosis in mice [13,16]. However, there are few studies on its application in bone tissue engineering materials for bone defect repair. IGF-1 has been shown to be a highly potent growth factor associated with bone regeneration, and our previous studies have shown that IGF-1 can induce osteogenic differentiation and the regeneration of skull defects [7,17]. In this study, both BBR- and DOPA-IGF-1@PLGA/HA microspheres had a positive effect on the repair of 5 mm diameter skull defects 60 days after implantation. The DOPA-IGF-1@PLGA/HA-BBR microspheres showed the best results, with the damaged areas almost completely covered. It has been shown that MC3T3-E1 cells expanded on microspheres had higher cell proliferation and osteogenic gene expression than those expanded on thin films [38]. In this study, in conjunction with previous in vitro experiments, the positive effect of the bioactive DOPA-IGF-1@PLGA/HA-BBR microspheres on osteoblast differentiation and bone regeneration in vivo was further validated. BBR-encapsulated DOPA-IGF-1@PLGA/HA microspheres can be used as osteogenic scaffolds to promote the filling of irregular bone defects and support the growth and adhesion of seed cells. In addition, BBR and DOPA-IGF-1 loaded in microspheres acted synergistically through the IGF-1R/PI3K/AKT/mTOR pathway to promote osteoblastic differentiation and mineralization and enhance bone defect repair. These results strongly demonstrated the potential of DOPA-IGF-1@PLGA/HA-BBR microspheres for clinical application.

## 3. Materials and Methods

### 3.1. Preparation of BBR-Encapsulated PLGA/HA Microspheres

Berberine chloride hydrate (Cat#B107342) was purchased from Aladdin, Shanghai, China. PLGA (lactide/glycoside ratio = 75/25, MW = 100,000), synthesized by the Changchun Institute of Applied Chemistry, Chinese Academy of Sciences (CIAC, Changchun, China). HA was purchased from Kejin Materials Technology Co., Ltd. (Guangzhou, China). The BBR-encapsulated PLGA/HA microspheres were prepared by electrified liquid jets and a phase-separation technique via a homemade device as in the previous report [10]. The device was composed of a high-voltage electrostatic generator (FRASER, Devon, UK), an extruder, and a receiving beaker. First, PLGA was dissolved in N-methyl pyrrolidone (NMP) to obtain a PLGA/NMP (7%, *w*/*v*) solution. Then, nHA and BBR were added to the PLGA solution. The mass ratio of PLGA: nHA was 9:1 (*w*/*w*), while the mass ratio of BBR added to the experimental group was varied (0.01 wt%, 0.05 wt%, 0.1 wt%, 0.5 wt%, 1 wt%). The BBR content is expressed as the mass ratio of BBR in the mixture solute (PLGA, nHA, and BBR). The solutions were magnetically stirred in the dark for at least 2 h until they were homogeneous. Then, the solutions were electrojetted into an ethanol aqueous solution with a concentration of 20%. The voltage was 8 kV, and the needle size was 28 G.

### 3.2. Immobilization of DOPA-IGF-1 on Microspheres

The expression and purification of recombinant IGF-1 bearing a Tyr-Lys-Tyr-Lys-Tyr tag at its C-terminal (YKYKY-IGF-1) were performed using our previously described method [23]. The dose of DOPA-IGF-1 was set at 100 ng/mL according to our previous report [7]. To convert YKYKY-IGF-1 to DOPA-IGF-1, the tyrosine hydroxylation was performed using tyrosinase (10U/μL, 1 μL) (Sigma Aldrich, Burlington, MA, USA), ascorbic acid (5 mg/mL, 500 μL, pH 7.2), YKYKY-IGF-1 (400 ng/mL, 500 μL, pH 7.2), and PBS (1 mL, pH 7.2). The mixed solution was incubated at room temperature for 2 h, sterilized with a sterile filter, and then adjusted to the set concentration. The solution was added to the sterilized microspheres and regulated to a pH of 8.5. Subsequently, the IGF-1-microsphere mixture solution was left to stand overnight at room temperature and washed three times with sterilized PBS.

### 3.3. Detection of BBR Encapsulation and Release Efficiency

In this part, the BBR concentration was detected at 270 nm with an ultraviolet spectrophotometer. First, the standard curve was plotted against the absorbance at 270 nm for different concentrations of BBR solution in methanol (Appendix A). The amount of BBR in the polymer solution used for microsphere preparation was denoted by M1. To detect the encapsulation efficiency (EE) and loading capacity (LC) of BBR, 100 mg microspheres was triturated and immersed in 1 mL of methanol to dissolve the BBR. The supernatant was collected by centrifugation. The mass of BBR remaining in the received solution was then determined as M2 calculated according to the standard curve. All data were obtained from three experiments.

EE and LC of the BBR-loaded microspheres were, respectively, calculated according to Equations (1) and (2):EE = M2/M1 × 100%(1)
LC = M2/100 mg × 100%(2)

The release behavior of BBR was investigated by immersing 100 mg microcarriers in 20 mL of PBS in a 20 mL tube fixed vertically in a shaker. This process is performed on a shaker with shaking of 80 rpm at 37 °C. After 0.5, 1, 2, 4, 8, 24, 48, 72, 96, 120, 144, and 168 h, 1 mL of PBS solution was collected and supplemented with the same volume of fresh PBS. The collected solution was extracted thrice with equal volumes of ethyl acetate, then evaporated using a rotary evaporator and re-solved in 1 mL of methanol. The amount of released BBR was determined as m calculated from the standard curve. The percentage of released BBR (RP) was calculated as follows:RP = (m1 + m2 + …… mn)/M1 × 100%(3)

### 3.4. Adhesion and Release of DOPA-IGF-1

The BBR-encapsulated PLGA/HA microspheres were placed in a 24-well plate, 100 ng/mL hydroxylated protein solution was added to 1 mL per well, and incubated at 4 °C overnight. The supernatant was taken out and stored as Sample 1 (S1). The unconjugated proteins were then extensively washed three times with 1 mL PBS containing 0.1% Tween 20 (pH = 7.2). The wash buffer of each time was also collected as samples (S2~S4). All samples were quantitatively detected by enzyme-linked immunosorbent assay (ELISA) using the Human IGF-1 DuoSet ELISA kit (R&D System, Minneapolis, MN, USA). The adhesion amount of fusion protein was calculated using the following formula:Adhesion Amount (AA) = Initial Amount (IA) − Sample Amount (SA)

IA was the total protein amount in the solution. SA was the total amount of unconjugated protein on the microspheres (SA = S1 + S2 + S3 + S4).

The release profile was determined by measuring the amount of IGF-1 remaining in the supernatant. The 24-well plate was plated with 5 mg DOPA-IGF-1-modified microspheres and 1 mL PBS (pH 7.2) per well. At specified time intervals (1 h, 2 h, 4 h, 12 h, 24 h, 48 h), 200 μL of supernatant was collected for analysis and replaced with fresh PBS (pH 7.2). The cumulative release ratio was calculated based on the total amount of protein obtained from the encapsulation yield.

### 3.5. Characterization of the Composite Microspheres

The general appearance of the microspheres was observed under an optical microscope. The microtopography of microspheres was captured under environmental scanning electron microscopy (ESEM, XL30 FEG, Philips, Hamburg, Germany). The particle size distribution of the microspheres was analyzed statistically using a granulometer (SALD-2300, Shimadzu, Kyoto, Japan). The longest and shortest diameters of the microspheres were measured using the ImageJ software (version 1.8.0), and the sphericity was defined as the ratio of the shortest diameter (SDM) and the longest diameter (LDM) of a microsphere:Sphericity = (SDM/LDM) × 100%

Fourier transform infrared spectroscopy analysis (FT-IR, Watford, UK) was performed to analyze the chemical composition of the microspheres.

### 3.6. Cell Culture

MC3T3-E1 cells (Shanghai Institutes for Biological Sciences, Chinese Academy of Sciences) were used in this research. The cells were expanded in Dulbecco’s minimum essential medium (DMEM, Corning, Manassas, VA, USA) containing 10% FBS (Gibco, GrandIsland, NY, USA), 100 mg/L streptomycin (Sigma, Burlington, MA, USA), and 63 mg/L penicillin (Sigma, Burlington, MA, USA). The third generation of cells was seeded on the microspheres. The proliferation of cells was detected by a CCK-8 assay kit (Solarbio, Beijing, China). The alkaline phosphatase activity of cells was detected by the alkaline phosphatase assay kit (ALP assay kit, Solarbio, Beijing, China). Briefly, the culture medium was discarded and the cells were washed twice with PBS. Then, 200 μL of cell lysis solution was added into each well, followed by freeze-thawing three times at −80 °C. After centrifugation at 12,000 rpm, the supernatant was collected. The ALP and total protein in the supernatant were detected by the ALP assay kit (Solarbio, Beijing, China) and the BCA assay kit (Thermo, Waltham, MA, USA), respectively. The control group was ALP activity of MC3T3-E1 cells grown in culture plates for 3 or 7 days without any treatment. The relative ALP activity was calculated as follows:Relative ALP activity= ALP (OD 405)/BCA (OD 562).

### 3.7. Reverse Transcription-Quantitative Polymerase Chain Reaction (RT-PCR)

Total RNA was extracted from cultured cells by Trizol extraction (Invitrogen, Carlsbad, CA, USA). The obtained RNA (1~5 μg) was used for cDNA synthesis with the Revert Aid First Strand cDNA Synthesis Kit (Thermo Scientific, # K1621, Waltham, MA, USA). The cDNA products were subjected to quantitative PCR (qPCR) using the qPCR SYBR Green Mix Kit (Stratagene, Santa Clara, CA, USA) with a real-time PCR System in 8 stripe optical tubes (Axygen, Manassas, VA, USA) in triplicate. The mRNA level was normalized to GAPDH using the 2^−∆∆Ct^ method. All the primers used in this study are listed in Appendix A. The specificity of the listed oligonucleotides was checked by Basic Local Alignment Search Tool (BLAST) against the homo RefSeq RNA database at NCBI.

### 3.8. Western Blot Analysis

Western blot analysis was performed as described as follows: cultured cells were washed twice with ice-cold 1× PBS and lysed with RIPA lysis buffer [50 mM Tris-HCl (pH 7.4), 150 mM NaCl, 1% (*v*/*v*) Triton X-100, 0.5% sodium deoxycholate, 0.1% SDS, and 1 mM EDTA] supplemented with protease and phosphatase inhibitor cocktails (Roche, Basel, Switzerland). The cell lysates were collected and centrifuged at 12,000× *g* for 20 min at 4 °C to remove debris and to obtain homogenates. The whole-cell homogenized extracts were fractionated by SDS-PAGE and transferred to a polyvinylidene fluoride membrane (Bio-Rad Laboratories, Hercules, CA, USA). Membranes were blocked with 5% BSA in TBST (10 mM Tris, 150 mM NaCl, and 0.5% Tween 20, pH 7.6) for 1 h then incubated with indicated antibodies overnight at 4 °C. Membranes were washed with TBST 4 times and incubated with HRP-conjugated secondary antibodies (1:5000~20,000) at room temperature for 1 h. Blots were washed 4 times with TBST and the target bands were detected with the ECL system (Bio-Rad, Hercules, CA, USA). The protein levels were normalized with respect to the loading control and analyzed using NIH ImageJ software. All the antibodies used in this study were listed in Appendix A.

### 3.9. Animals and Experimental Protocol

Ten female Wistar rats (8 weeks, 200–240 g) were employed in this experiment. All animal experiments were approved by the Institutional Animal Care and Use Committee of Changchun Institute of Applied Chemistry, CAS (NO. 2201042839670, approval code: 20230066). After being anesthetized with an intraperitoneal injection of 3% sodium pentobarbital (30 mg/kg), the animals were immobilized on the operating board. After skin preservation and sterilization, a posterior incision of approximately 20 mm in length was made along the centerline from the midpoint of the two posterior canthi. To fully expose the cranial parietal bone, the subcutaneous tissue was bluntly separated. A dental drill was used to bore two circular defects 5 mm in diameter on each side of the parietal bone to the depth of the dura mater. The 20 defects in the ten rats were evenly divided into five groups. Different samples were implanted in the defect region and the non-implantation group was used as the blank control group. Samples from the same group were assigned to different rats. A total of 80,000 IU/day of penicillin sodium was administered intramuscularly for 5 days following surgery.

Micro-CT (Perkin Elmer, QuantumGX2, Waltham, MA, USA) was performed every 20 days to observe the effect of repair during the process. The regeneration coverage rate was detected by ImageJ. The bone volume fraction (BVF, BV/TV) at the defect site was detected by CTAn. The final repair effect was evaluated by tissue section staining (H&E and Masson, Solarbio, Beijing, China).

### 3.10. Statistical Analysis

Unless otherwise stated, the measurements were performed in three or four independent replicates. The obtained data are expressed as the mean ± SD deviation. All immunoblotting data were quantified by NIH ImageJ software. For multiple comparisons of three or more sets of samples, one-way ANOVA or two-way ANOVA tests were used. The differences were considered statistically significant for *p* < 0.05 (* *p* < 0.05, ** *p* < 0.01, *** *p* < 0.001).

## 4. Conclusions

In this study, a novel BBR-encapsulated PLGA/HA microsphere was fabricated by electrified liquid jet and a phase-separation technique, followed by modification with DOPA-IGF-1. The BBR encapsulated in the composite microsphere can be released sustainably for more than 168 h. DOPA-IGF-1 was bound to the composite microsphere with high immobilization efficiency and stability. The composite microsphere showed good biocompatibility due to the addition of nHA and IGF-1. BBR and IGF-1 can synergically promote osteogenic differentiation and bone matrix secretion in MC3T3-E1 cells in vitro. The underlying mechanisms were further clarified. The BBR can effectively activate the IGF-1R/PI3K/AKT/mTOR pathway to enhance IGF-1-mediated osteogenic differentiation. The in vivo results indicated that the IGF-1@PLGA/HA-BBR microspheres could effectively promote the complete bony connection covering the defect site and enhance the effect of bone defect repair. All the results confirmed that the IGF-1@PLGA/HA-BBR composite microsphere with good flowability, adjustable particle size, and well-defined 3D structure was promising to be implanted into irregular bone defects for clinical applications. The slow release and synergistic function of BBR and IGF-1 can promote the osteogenic differentiation of cells and the secretion of bone matrix.

## Data Availability

The datasets generated and/or analyzed during the current study are included in this published article and its Appendix A.

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
