# Peer review of "Berberine-Encapsulated Poly(lactic-co-glycolic acid)–Hydroxyapatite (PLGA/HA) Microspheres Synergistically Promote Bone Regeneration with DOPA-IGF-1 via the IGF-1R/PI3K/AKT/mTOR Pathway"

_ijms, 2023, doi:10.3390/ijms242015403_

Round 1
Reviewer 1 Report
Although this is a well set up experiment, the authors need to highlight its importance and innovative characteristics.
1) Please describe in detail microsphere [size and orientation].
2) Considering ALP, authors use it as early or late marker for differentiation?
3) Was the increase of ALP expected, at day 3 and 7? Please compare your results with other papers.
4) For what kind of damages, these microspheres were recommended.
5) Which is the innovation part of this experiment? Please highlight the differences and the superiority of your experimental model compared with others similar protocols reported.
Reviewer 2 Report
The manuscript entitled "Berberine - encapsulated PLGA/HA microspheres synergistically promote bone regeneration with DOPA-IGF-1 via IGF-1R/PI3K/AKT/mTOR pathway" is an interesting work dealing with the preparation, in vitro and in vivo evaluation of microparticles intended to improve bone regeneration. Besides, studies were performed in order to determine the mechanism of the observed effects. Nevertheless, I found some flaws that would require a revision by the authors. Please, see below:
1. The introduction would benefit from further literature data to support the choice of excipients and active ingredients.
2. In my opinion, the listing of so many numbers in the narrative text (EE, LC, PR) makes it hard to capture the whole picture of the results. I recommend the addition of some tables which may present the numerical data for different experiments.
3. Composition and coding of the different investigated batches would also provide better clarity. Currently, the amount of BBR per formulation is unclear.
4. The Materials and Methods section would need some modification - the materials type and origin is not given (there are different molecular weight (PLGA, HA, etc.);
5. How was the in vitro release conducted? It is unclear if any agitation was applied. Do the microspheres float? Was a dialysis membrane used?
6. Why the formulation containing 0.01% BBR releases more completely than 0.5 and 1BBR? Please provide proper discussion.
7. How was the effect of BBR content on MCT3T-E1 evaluated? It is unclear whether different concentrations of the tested microspheres with definite BBR content are evaluated or just different microspheres based on the BBR content? The release behavior shows significant difference in the release profiles and therefore it can cause also significant differences in the biocompatibility? Why free BBR and non-loaded microspheres were not investigated? Proper discussion of the observed effects is also advisable.
8. What was the control group for ALP activity?
9. The data regarding section 2.8 Repair effect of Skull injury should also be provided in table for easier and more clear comparison. Proper discussion is needed.
10. Abbreviations should appear the first time being mentioned. Please, revise the manuscript.
There are some English corrections needed- some unclear sentences appear in text: e.g. line 62; line 164; line 225-226;
There are some incorrect verbs: line 237 was instead of were;
Round 2
Reviewer 1 Report
Thank you for following all the suggestions and answering my questions one by one.
Reviewer 2 Report
The revised version of the manuscript has addressed adequately my recommendations. Its scientific value has been improved and it would contribute to the knowledge on the topic.
One minor suggestion is to omit most of the numerical values in the text. The break the text flow and when they are presented in a table I think it is not necessary to repeat it.
